# Control of complex systems with generalized embedding and empirical dynamic modeling

**Joseph Park**[1,2]*, **George Sugihara**[1], **Gerald Pao**[2]

**1** Scripps Institution of Oceanography, University of California San Diego, La Jolla, CA, United States of America, **2** Biological Nonlinear Dynamics Data Science Unit, Okinawa Institute of Science and Technology, Onna-son, Okinawa, Japan

* JosephPark@IEEE.org

**Data Availability Statement:** All files are available in the Zenodo archive: Joseph Park. 2023. State based dynamic intervention. https://doi.org/10.5281/zenodo.8408617.

## Abstract

Effective control requires knowledge of the process dynamics to guide the system toward desired states. In many control applications this knowledge is expressed mathematically or through data–driven models, however, as complexity grows obtaining a satisfactory mathematical representation is increasingly difficult. Further, many data–driven approaches consist of abstract internal representations that may have no obvious connection to the underlying dynamics and control, or, require extensive model design and training. Here, we remove these constraints by demonstrating model predictive control from generalized state space embedding of the process dynamics providing a data–driven, explainable method for control of nonlinear, complex systems. Generalized embedding and model predictive control are demonstrated on nonlinear dynamics generated by an agent based model of 1200 interacting agents. The method is generally applicable to any type of controller and dynamic system representable in a state space.

## 1 Introduction

Complexity confounds control, yet control is an essential element of many complex systems wherein regulatory feedback emerges for targeted behaviors, for example, the citric acid cycle, mRNA translation and biologic homeostasis. Control theory arose from the bedrock of linear time-invariant (LTI) theory where complexity is reduced to independent components relying on Laplace domain analysis of system stability and controllability through the transfer function [1]. However, nonlinearity and complexity are inherent in physical and cybernetic systems leading modern control theory to embrace model predictive control (MPC) where system dynamics are encapsulated in a process model predicting future states to inform regulatory feedback [2].

In simple systems the dynamics can be expressed with equations, for example, the Kalman filter where equation–based states are informed with Bayesian estimates of measurement and process uncertainty. In complex systems it can be a challenge to mathematically express the dynamics in which case data–driven models provide an alternative. For example, in fuzzy controllers the dynamics are abstracted in a rule base requiring expert knowledge of the system

**Funding:** The author(s) received no specific funding for this work.

**Competing interests:** The authors have declared that no competing interests exist.

[3], while machine learning applications such as neural networks rely on training data and model parameterization to encode the dynamics [4]. A concern with many machine learning models is the barrier of mechanistic opaqueness: how does one relate internal states of the model to explanatory insight? Additional concerns include overfitting, limited generalization, and, resource intensive training and tuning.

Here we demonstrate the use of generalized state space embedding [5] as a data–driven process model representation and leverage empirical dynamic modeling (EDM) [6, 7] as a nonlinear predictor applied to complex system control. EDM is a state space nearest neighbor projector using either a state space simplex [8] or localized state space kernel regressor [9] to predict future states. EDM has demonstrated broad applicability across multiple disciplines to characterize, forecast, and identify intervariable dependencies and causal links of complex systems [6, 7]. A schematic depiction of EDM prediction is shown in Fig 1.

Advantages of generalized state space embedding and EDM include

1. The process model is data–driven and encapsulated in the dynamical state space

2. Information flow from observations to predictions is deterministic and traceable

3. Extensive parameterization or training are not required.

Before detailing the method and application we review model predictive control applied to chaotic systems, networked dynamical systems, time delay embedding and dynamic mode decomposition (DMD), as well as state–of–the–art multiagent and adaptive control with neural networks.

## 1.1 Chaos control

Application of control to chaotic dynamics was pioneered by Ott, Grebogi, and Yorke, commonly known as the the OGY method [10]. It is predicated on the observation that a chaotic attractor can be considered to have to a large number of attracting periodic trajectories such that judiciously applying small, time-dependent perturbations to an available system parameter can nudge the dynamics to pursue a desired orbit. The method entails identification and selection of specific orbits to be controlled, followed by linearization and application of a linear state feedback controller to nudge the dynamics at predefined critical points into a stable orbit. The selection of specific control points, orbits, and linearization may be problematic in complex systems and networks with multiple interactions and incomplete or noisy observations. As an alternative, empirical dynamic modeling offers techniques that are fully nonlinear and effectively deal with multiple interactions in real world, noisy data.

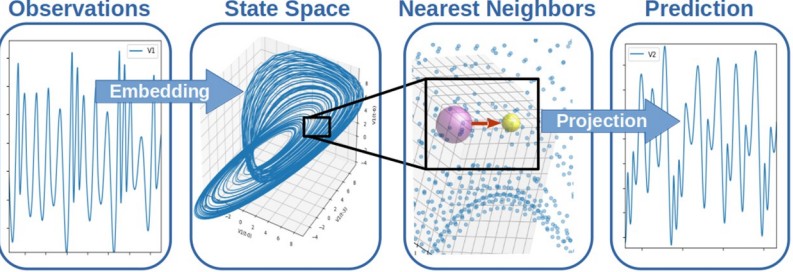

**Fig 1. Schematic of empirical dynamic modeling (EDM) processing.**

## 1.2 Network control

Although feedback control and cybernetics *are* disciplines of interacting components, additional complication arises when the system itself is comprised of a network of systems. Accordingly, analysis and techniques for control of complex networked systems is an active discipline [11–13].

Regarding the more general control of complex networks, Liu et al. [14] converged graph theory with control theory finding that sparse inhomogeneous networks, which emerge in many real complex systems, are the most difficult to control requiring a relatively high number of control nodes. Conversely, dense and homogeneous networks can be controlled using relatively few control nodes. Continuing this avenue, Liu et al. [15] proposed graph analysis of complex networks identifying a lower bound on the number of strongly connected nodes (defined as the largest subgraphs maintaining a directed path from each node to every other node in the subgraph) reflecting the minimum number of sensors (a selected subset of state variables from which one can determine all other state variables) required to observe the system dynamics. This suggests network topology influences the controllability of complex network systems, although, the intrinsic dynamics alone impose fundamental constraints on complex system controllability [16].

A comprehensive review of complex system control is provided by Liu and Barabási [13] suggesting several fundamentals:

1.  Topologies of most real systems share numerous universal characteristics

2.  These universal topological features result from common dynamical principles governing their emergence and growth

3.  Topology fundamentally affects the dynamical processes taking place on the network

4.  Network topology of a system affects the ability to control it.

Although topology impacts and informs system control, altering network structure to effect or enable control may not be possible. Here, we deem the state space representation dictated by observations as an invariant, seeking to regulate dynamic trajectories to avoid undesired states. Trajectory regulation can be achieved with either *state variables* and/or external *system variables* since generalized embedding and EDM prediction can incorporate both in the state space.

## 1.3 Dynamic mode decomposition

Dynamic mode decomposition (DMD) offers a data–driven method to estimate eigenfunctions of the Koopman operator, a linear operator describing evolution of scalar functions (observables) along trajectories of a nonlinear dynamical system. Linearization is attractive from a control perspective as it directly translates to the rich and highly successful LTI control theory.

Theoretically, DMD represents a finite–dimensional nonlinear system with a set of linear operators acting in an infinite-dimensional space. One therefore expects some form of optimization is needed to converge a solution. Further, systems of high complexity can be problematic for Koopman decomposition thereby requiring some form of regularization [17]. Extended DMD (EDMD) casts the eigenvalue problem onto a *dictionary* of observables (basis functions) spanning the finite dimensional subspace on which the Koopman operator is approximated. While this can improve solution accuracy and convergence, it adds complexity and dependence on the careful selection of dictionaries. To address this, hybrid deep–learning

DMD applications identifying suitable observables have emerged, but at the cost of increased numerical load, parameterization, and algorithmic complexity [18–20].

Even with these representational and computational burdens, the transformation of a non-linear system to a linear basis offers a path to leverage the well–understood machinery of LTI control theory to nonlinear systems. The seminal contribution of Korda and Mezić [21] actual-izes this with application of the Koopman operator to a state space synthesis (product) of the uncontrolled dynamics and that of all available control actions. The method entails a nonlinear transformation of the data to a higher–dimensional space (termed *lifting*) enabling linear decomposition in the lifted space. A similar prescription was demonstrated by Proctor et al. [22] where the measurement and control data are linearly combined, there however, the decomposition is performed on the data rather than in the linearized, lifted space.

As successful as these method are, they nonetheless require optimization of the observation functions into the higher–dimensional space, and rely on the premise that a linearized control action applies generally to nonlinear system control.

### 1.4 Sparse identification of nonlinear dynamics

The sparse identification of nonlinear dynamics (SINDY) framework can be considered a gen-eralization of DMD that has been adapted to model predictive control [23]. Similar to the EDMD dictionary, SINDY requires specification of a *library* of nonlinear functions that encompass the underlying dynamics. Regression with a sparsity constraint is used to select library functions that best represent the dynamics. As is the case for the EDMD dictionary, careful specification of the candidate library is crucial.

### 1.5 Artificial neural network

Owing to their universal approximation capability neural networks were quickly adopted to model process control in the previous century [24]. Since then, neural network and hybrid neural network models and controllers continue to evolve and succeed in modeling and con-trol of nonlinear dynamical systems. Recent advances allow these systems to learn and control unknown nonlinear dynamics, for example, using radial basis functions [25], recurrent wavelet neural networks [26], and adaptive control with recurrent neural networks [27]. We note a dis-tinction of generalized embedding and state space projection is that network design, parame-terization, and training are not needed. Further, the direct connection between observation functions and the generalized embedding means there is a deterministic, traceable chain of information from observations to model predictions, a feature that is commonly absent from neural network models.

### 1.6 Multiagent control

Regarding distributed control of multi–agent systems with potentially unknown dynamics, Shahvali et al. [28] propose a dynamic event-triggered control framework for neural network controllers. The method provides a consensus of heterogeneous multi-agents utilizing a mini-mal learning parameter technique to avoid updating the neural network weight vectors. The dynamic event-triggered control scheme effectively limits unbounded behavior while reducing control energy.

### 1.7 Fractional order control

Fractional calculus is a well–established generalization of integer–order calculus with application to many real–world dynamical systems. Fractional–order dynamical systems can be

modeled with a fractional differential equation using derivatives of non-integer order amenable to systems with power–law dynamics and thus heavy–tailed distributions. Application of fractional calculus to control has resulted in fractional order control (FOC) applicable to non-linear dynamical systems [29]. Recent advances have developed event–driven control for systems with unknown dynamics while minimizing the impact of faults and reducing control action of neural network controllers [30, 31]. It can be noted these methods have non-trivial complexity as the error surface, adaptive control laws, and fault compensator are computed and updated at each time step, steps that are not required in the proposed generalized embedding approach.

### 1.8 Kalman–Takens filter

A step toward generalized embedding for model predictive control was taken with the Kalman–Takens filter [32] where the canonical Kalman filter state and observation functions are replaced with a Takens time delay embedding and nearest neighbor projection. As noted above, this has the advantage of not requiring explicit mathematical model of the dynamics and observations, however, the Kalman filter as a linearized predictor is but one of many possible prediction models, and for complex, multivariate systems, there is information to be leveraged with generalized, multivariate embedding as discussed in the following section.

## 2 Generalized state space and empirical dynamic modeling

Following recognition of low–dimensional determinism in the 1960's, impressive gains to unravel deterministic dynamics through time delay embedding have been made. A classic exposition is given by Sauer et al. [33], with a recent overview by Tan et al. [34].

Deyle et al. [5] considered the remarkable work of Takens, Packard, Crutchfield, Sauer and others [33, 35, 36] to arrive at equally remarkable statements of generalized state space embeddings where time series delay coordinates, and/or, suitable combinations of observation functions represent system dynamics in mathematically consistent embeddings. When coupled with empirical dynamic modeling, generalized embeddings provide a powerful basis for unraveling complexities of nonlinear, multivariate systems.

To generalize the use of state space embeddings as the basis of model predictive control we apply empirical dynamic modeling (EDM) to predict state variables. We note several distinctions from methods discussed above:

1. The data defines the state space. There is no *a-priori* specification of candidate library functions or dictionary

2. Predictions are made directly in the state space

3. Direct interpretability is maintained since there is no external representation, such as a deep-learning model or functional abstractions

4. Recursive optimization is not needed

5. Methods to assess cross variable interactions in the dynamics are inherent in EDM facilitating direct inspection of multivariate dependencies of the dynamics and control.

### 2.1 Intervariable dependencies

EDM operates in the multidimensional state space of the system dynamics enabling *cross mapping*, the prediction of a state variable from one, or, several other state variables. Cross

mapping culminates in *convergent cross mapping*, a state space method to assess causal inference between variables [37]. The ability to cross map between variables, and therefore predict one variable from another as an indication of shared information and causal states can be highly informative toward mechanistic understanding of complex, multivariate systems.

Here, we employ cross mapping with EDM sequential locally weighted global linear maps (s–map) [9] to predict a feedback control variable. S–map is a forerunner and generalized cousin of the popular locally linear embedding (LLE) widely used in machine learning [38]. Instead of using a fixed number (k) of state space nearest neighbors upon which a local linearization is performed, s–map localizes state space neighbors with an exponential decay kernel. The width of the kernel, and thus the extent of locality, is controlled with a tunable parameter $\theta$ to best represent the degree of nonlinearity. Importantly, s–map intervariable Jacobians quantify nonlinear time–varying interactions between state space variables [39]. We capitalize this feature to infer dynamical dependencies of the state space and control variable. The s–map algorithm is detailed in Appendix.

## 3 Materials and methods

### 3.1 Model predictive control with EDM

The innovation demonstrated here is application of generalized state space embedding and empirical dynamic modeling as a process model for model predictive control. A schematic representation of the architecture is depicted in Fig 2 with application details provided in following sections.

### 3.2 Nonlinear dynamics: A model of civil disobedience

To demonstrate the method on a complex, dynamic network, we employ an agent–based model (ABM) from Epstein generating time series of civil disobedience [40]. However, the EDM based model predictive control (MPC) applies to any dynamical system.

Epstein's model simulates dynamics of civil disobedience among a population of citizens under rule of a central authority enacting law enforcement and incarceration. Citizens maintain a level of risk aversion to avoid arrest, balanced against grievance toward the government based on perceived government legitimacy [40]. Agents interact with other agents and the

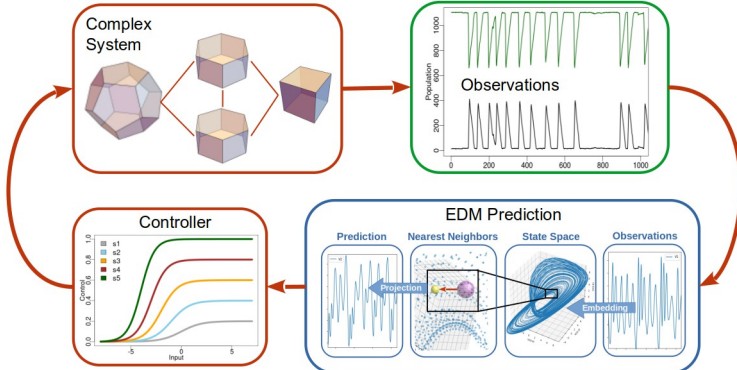

**Fig 2. Schematic of state based control.** A complex system generates time series observations modeled in state space. State space predictions inform a controller modulating a system or state variable. The predictions do not require an explicit mathematical or structural model.

environment to produce complex nonlinear dynamics, for example, the waiting time between episodes of rebellion follow an exponential distribution, a hallmark of punctuated equilibrium.

Our model is a slight variation of Epstein's model, although we maintain the world domain of 1600 cells and population of 1200 interacting agents. In both models citizens decide whether to rebel by assessing their grievance toward the government against the risk of being arrested. If their grievance outweighs the risk of arrest, assessed numerically in the Epstein model as the exceedance of a static threshold, the citizen becomes Active (rebels). The Epstein model also employs a static value of government legitimacy.

Using static legitimacy and a static grievance threshold, our model demonstrates the punctuated equilibrium of Epstein's model. Our model then allows both legitimacy and the grievance threshold to be variable. Legitimacy is randomly varied in time and magnitude, and we cast the grievance threshold as a government controlled variable termed *propaganda*.

The government uses propaganda to influence citizen decisions to rebel. Nominal population dynamics exhibit quiescent periods with large populations of Quiet citizens and low numbers of Active (rebelling) citizens, punctuated by episodes of civil disobedience brought under control by law enforcement. We then unbalance the dynamics by varying government legitimacy which can result in sustained rebellion. To restore order the government can respond with increased propaganda to reduce citizen perceived grievance reducing the number of citizens deciding to become Active.

Government decisions to regulate propaganda are based on a logistic controller transforming the predicted number of Active citizens into a value of propaganda effort. The controller is described in Appendix. The Active forecast is based on a state space created from the number of Quiet and Jailed citizens. State space cross mapping is performed with the Python package pyEDM [41]. The model is deployed in NetLogo [42] with the model interface and nominal population time series shown in Fig 3. Details of the model are provided in Appendix.

## 3.3 State space from generalized embedding

EDM operates in the state space of the dynamics, it is therefore important the state space represent the dynamics as completely as possible. Theoretically, this is achieved with satisfaction of the Whitney embedding theorem [43] ensuring the dimensional expansion from observation functions to state space dimension is adequate. One way to assess this is by examining how well state spaces with different embedding dimensions represent (predict) the observed dynamics. Results of this evaluation on a univariate time series of the number of Active citizens from a model run are shown in Fig 4 indicating an embedding dimension of $E = 5$ provides a useful lower bound. The high forecast skill and low embedding dimensions at forecast intervals $T_p = 1, 2$ shown in Fig 4 are a reflection of serial (auto) correlation. To avoid influences of serial correlation model predictions are made at a forecast interval of $T_p = 5$.

The state space consists of a 6 dimensional feature vector (exceeding the 5 dimensional lower limit suggested in Fig 4). The embedding contains the Jailed and Quiet variables, and, their $t + 2$ and $t + 4$ time shifted values, all from past observations. To create the initial state space from which predictions are made there is a model warm-up period where the ABM produces values of the Quiet and Jailed variables, but no controller estimates are made until there have been 3000 Active states recorded.

## 3.4 Process model prediction

At each time step s–map predicts the number of Active citizens from the state space of Jailed and Quiet observables. The forecast number of Active citizens is input to the controller adjusting the level of propaganda.

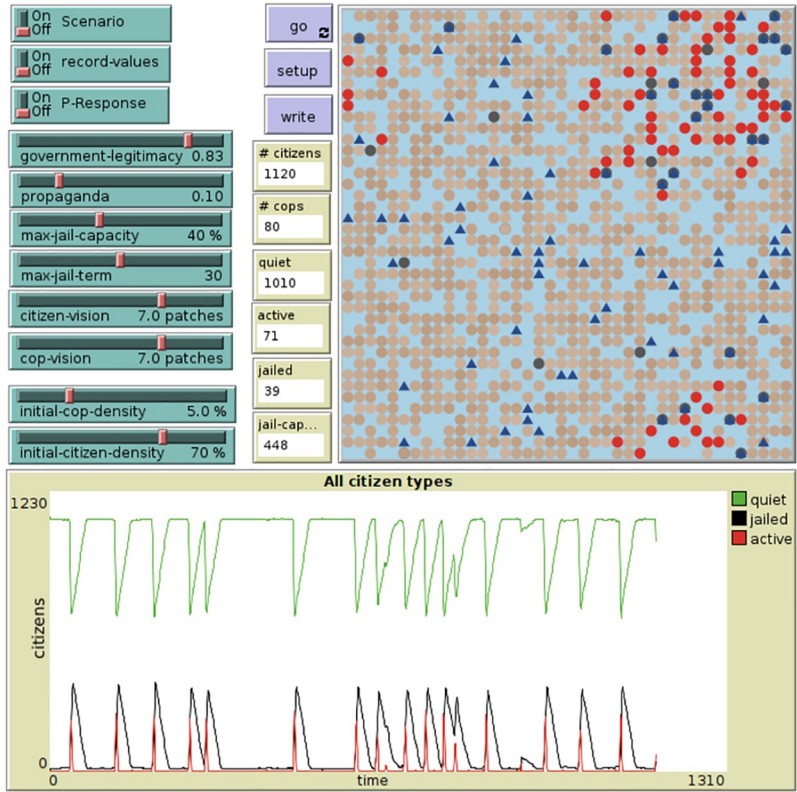

**Fig 3. Agent-based model interface (top) and nominal results (bottom) for Epstiens model of civil disobedience with constant legitimacy and propaganda.**

## 4 Results

Fig 5 shows model results under scenarios of randomly varied legitimacy. With no control the system enters a trapped state of sustained rebellion, whereas with control, the dynamics are regulated avoiding the trapped state. The controller is robust, preventing

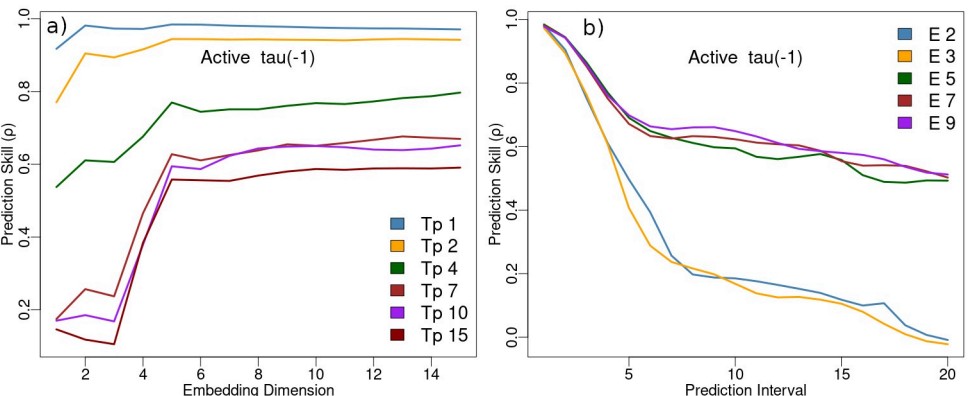

**Fig 4.** a) EDM simplex out–of–sample prediction skill (Pearson $\rho$) of the agent–based model Active variable as a function of state space embedding dimension at different forecast intervals ($T_p$). b) EDM simplex out–of–sample prediction skill (Pearson $\rho$) of the Active variable as a function of forecast interval at different state space embedding dimensions (E).

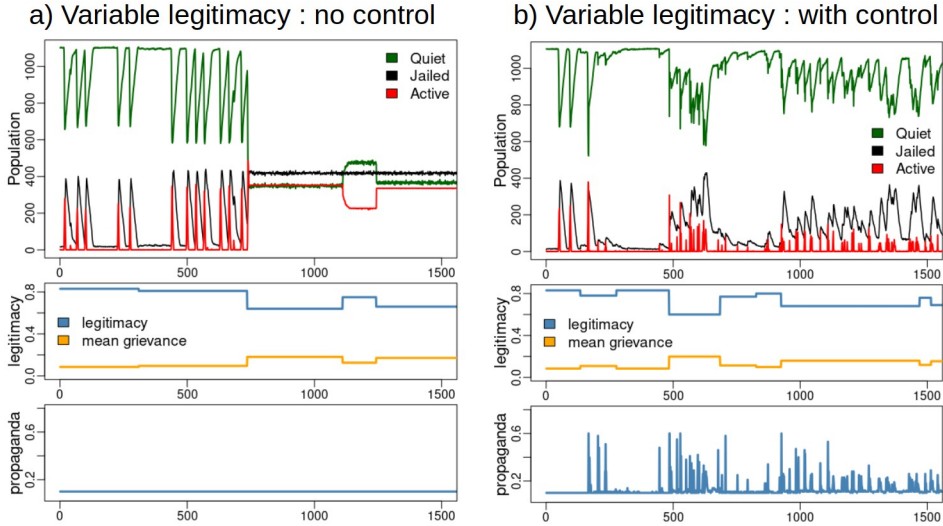

**Fig 5.** a) Civil disobedience model output under a scenario of variable legitimacy and no control. Punctuated equilibrium dynamics transition to a trapped state of sustained rebellion. b) Model output under a scenario with variable legitimacy and propaganda control. The control prevents the trapped states and sustained rebellion.

emergence of rebellion in all model runs attempted (more than 100 runs), and stable (see Appendix).

To examine system states and their trajectories under nominal (constant legitimacy), non-controlled, and, controlled (random legitimacy) conditions, we plot the 3-D space of variables Active, Quiet and Jailed in Fig 6. This view of the dynamics finds emergence of the trapped states in the non–controlled scenario, and, demonstration that the controller redirects state trajectories avoiding sustained rebellion.

Next, we use s–map state space regression coefficients to examine the relation between Active states and propaganda encapsulated in the $\partial \text{Active}/\partial \text{propaganda}$ terms under regimes of low and high legitimacy. Fig 7 plots the distribution of $\partial \text{Active}/\partial \text{propaganda}$ variance partitioned by states with legitimacy below, and above 0.7 indicating that during states of high legitimacy changes in Active states with respect to propaganda are not highly variable. Conversely, during states of low legitimacy it appears the impact of propaganda in determining Active states is highly variable, leading to an inference that propaganda is more effective as a population control when grievances against the authority are minimal.

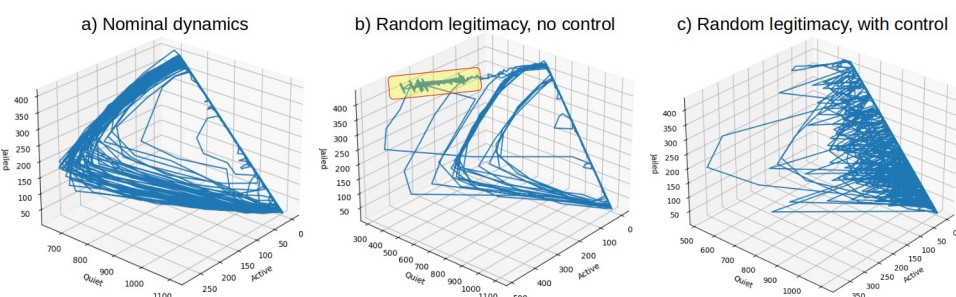

**Fig 6. 3-D projections of state variables Active, Quiet and Jailed.** a) Nominal dynamics with constant legitimacy and no control. b) Random legitimacy with no control. Trapped states are highlighted. c) Random legitimacy with control.

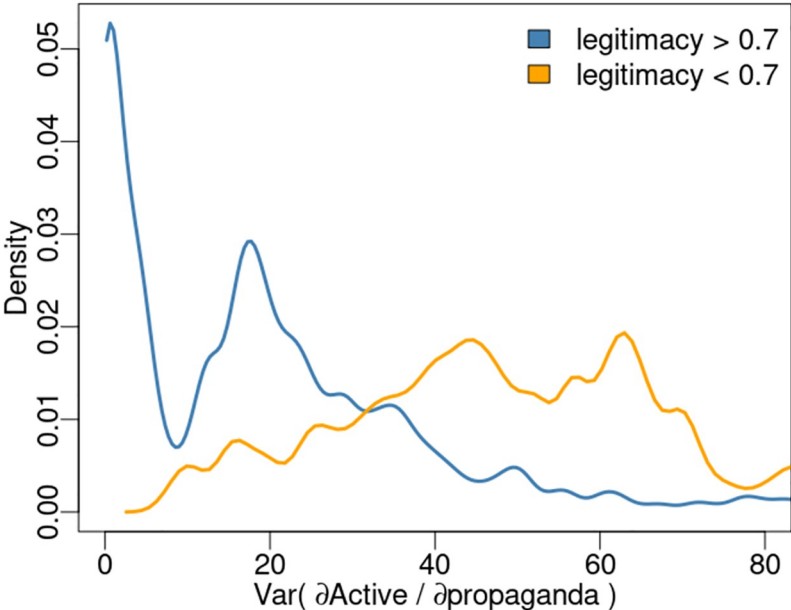

**Fig 7. Probability density estimates of s–map state space coefficient ∂Active/∂propaganda variance under conditions of low (< 0.7) and high (> 0.7) legitimacy.**

## 4.1 Model comparison

The proposed method uses generalized embedding and empirical dynamic modeling (EDM) to predict states informing a controller. To compare model predictive skill against other data–driven models we compare model performance to dynamic mode decomposition (DMD) and a neural network (ANN). All models are compared on the same data, a 6 dimensional generalized embedding from 3100 observations generated by the NetLogo model. The first 1500 observations are used as the training set, while observation rows 1601–3100 constitute the out–of–sample test set as shown in Fig 8. The generalized embedding is created as specified in section 3.3.

The DMD model is computed using higher order DMD (HODMD) from the python package pyDMD [44]. HODMD can perform DMD on time series data using time-lagged snapshots through superimposed DMD in a sliding window [45]. The snapshot matrix is the generalized embedding 6 dimensional data created from the data shown in Fig 8 as detailed in section 3.3.

The neural network model is created using the R package neuralnet [46]. Multiple network configurations were tried including networks with 1, 2, or 3 hidden layers. The best predictive skill was found with a network of 6 input nodes, one for each feature vector time series, two hidden layers with 3 nodes each, and a single node output layer as shown in Fig 9.

A comparison of model results is listed in Table 1. The EDM S-map model performed well with an out–of–sample prediction correlation coefficient of 0.98. The DMD model was not able to provide useful prediction of the number of Active agents, which is not surprising as DMD is based on a linear time step regression well-suited to problems with smoothly varying dynamics. Here, the abrupt Poisson distributed Active events do not contain prior information suitable for DMD [47, 48].

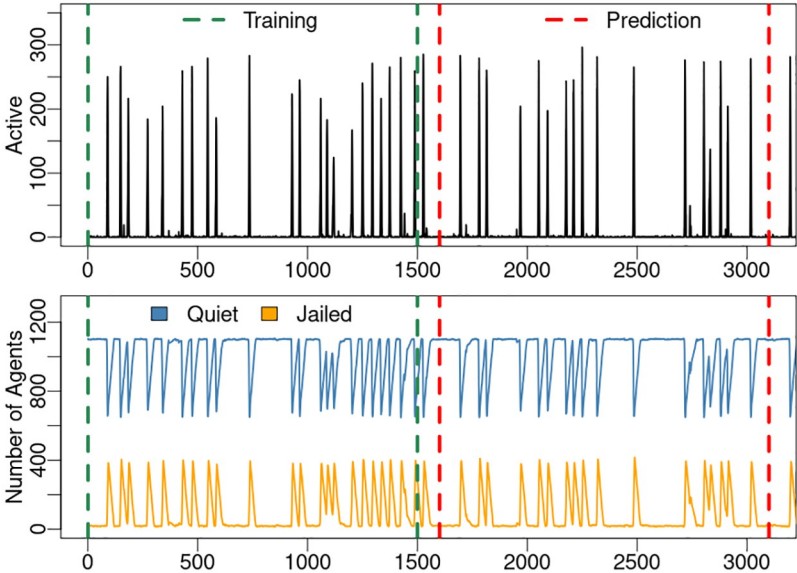

**Fig 8. Data used for model comparison.** Top: Predicted variable: number of Active agents. Bottom: Observation variables: number of Jailed and Quiet agents. The state space is created from the observation variables as detailed in section 3.3. All models are created on the training set consisting of observation time steps 1–1500, and evaluated on the out–of–sample prediction set taken from observations time steps 1601-3100.

The neural network model performed well predicting Active event initiation times, but failed to capture amplitudes of the events resulting in a best out–of–sample prediction correlation coefficient of 0.34 over multiple network configurations and initializations. With additional effort a neural network with an optimum architecture and initialization can be found to accurately predict the Active states, however, an advantage of generalized embedding and EDM prediction is that no such design and optimization are needed.

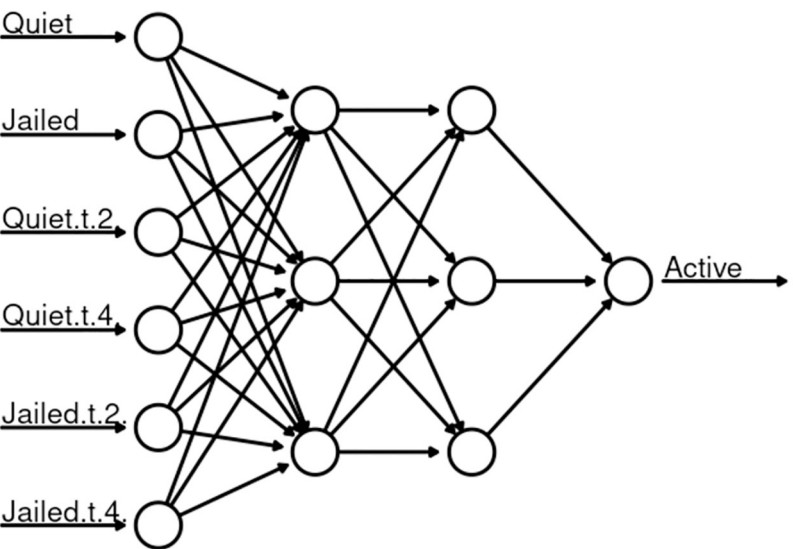

**Fig 9. Neural network to predict the number of Active agents.**

**Table 1. Model comparison.**

| Model | Skill ($\rho$) | Design | Training |
|-------|------|--------|----------|
| EDM S-map | 0.98 | No | No |
| DMD | 0.00 | No | No |
| ANN | 0.34 | Yes | Yes |

Comparison of models for out–of–sample prediction of Active states. Skill ($\rho$) is the correlation coefficient of model predictions on the withheld data. Design indicates whether the model architecture has to be designed with a network definition or prescription of model functions. Training indicates whether model training or optimization is required.

# 5 Conclusion

Nature finds solutions to control and regulation of complex systems through natural selection, a feedback control process. Interestingly, natural systems are data–driven systems, a feature we emulate in this work. As we seek to unravel complexities converged by nature or technology into the control of complex systems, we are faced with the difficulty that process models are not always tractable. Here, we demonstrate the use of data–driven, generalized state space prediction as a method to generate state estimates of process dynamics for model predictive control. Several advantages are provided with this approach:

1. The process model state space is constructed from observation functions without the need for model design, detailed parameterization, optimization, or training. This can be a significant simplification compared to complex equations, decomposition methods, or neural networks.

2. Model predictions are made directly in the state space.

3. State space kernel regression with sequential globally weighted local linear maps (s–map) allows quantification of interactions between dynamical variables with direct relevance to control and understanding of system dynamics.

To generate nonlinear dynamics of a complex system where the underlying behaviors are understood but there does not exist an explicit mathematical formulation, we employ an agent based model of 1200 interacting agents which collapses to an undesired fixed state in the absence control. The proposed method is able to accurately predict states of the system, control the system, and quantify the impact of propaganda on citizen decisions to rebel (become Active). As such, we infer propaganda is more effective as a population control when grievances against authority are reduced.

This work demonstrates that synthesis of agent–based models with state space prediction can be leveraged to explore the impact of dynamic interventions, shown here by quantifying the relationship between propaganda and citizen behavior. The ability to refine and reformulate the agent–based model in response to control variables provides a way to probe the impact of complex system interactions and interventions under scenario–based assessments.

Although we use an agent–based model as a generator of complex system dynamics, the proposed method is not limited to the use of an agent–based model or a specific type of controller. The method is generally applicable to multivariate, complex systems. Additionally, generalized embedding and EDM cross mapping lead directly to mechanistic understanding of system interactions. As such, we anticipate the method can provide a useful contribution to the toolbox of model predictive control of nonlinear complex systems.

## 6 Appendix

### 6.1 NetLogo model

The NetLogo model was run with NetLogo version 6.3.0 [42]. EDM state space predictions were run with version 1.14.3 of the Python pyEDM package [41] through the NetLogo `py` extension. The model is available at https://doi.org/10.5281/zenodo.8408616 [49]. We use the NetLogo convention for variable names with interspersed dashes. For example, in `risk-aversion` the character "`-`" does not indicate subtraction, but part of the variable named `risk-aversion`.

**6.1.1 Agents.** The model has two agent types: *Citizens* and *Cops*. The central authority (government) is represented as a collection of parameters and rules influencing agent behaviors. The primary government variable is *legitimacy*. In Epstein's model it is a constant parameter against which agent dynamics unfold. In our control scenarios it is a random variate.

Citizens can be in one of three states:

1. Quiet

2. Active

3. Jailed

Quiet citizens can decide to transform to the Active state if their level of `grievance` against the government exceeds a threshold reflecting the current state of government `propaganda`. Higher levels of `propaganda` raise the threshold by which citizens decide to become Active. The decision is also influenced by individual `risk-aversion` and `arrest-probability`, detailed below. Individual citizen `grievance` is proportional to their `perceived-hardship` (defined below) and inversely proportional to government `legitimacy`.

The transition from Quiet to Active, or Active to Quiet, is dynamically assessed for each citizen at each model time step. When an Active citizen is arrested they transition to the Jailed state. Upon completion of the `jail-term`, a random variate drawn at arrest time, the citizen is returned to the Quiet state.

Cops search for Active citizens. If there are Active citizens within the cops' `vision` (cell radius, defined below), and `jail-capacity` is not exceeded, one Active citizen is randomly selected and Jailed.

Each citizen maintains the following internal parameters:

1. `risk-aversion`: Random deviate in U[0, 1]

2. `perceived-hardship`: Random deviate in U[0, 1]

3. `active?`: True if Active

4. `jail-term`: Random deviate in U[1, max- jail- term]

5. `vision`: cell radius for movement and arrest probability

**6.1.2 Behaviors.** The fundamental behaviors are:

- Citizens

  1. `move-citizen`

  2. `citizen-behavior`

- Cops

    1. `move-cop`

    2. `enforce`

- Government

    1. `predict Active`

    2. `set propaganda`

The `move-citizen` and `move-cop` behaviors move the agent to a new cell within the agent's `vision` radius. The `citizen-behavior` decides whether the citizen transitions to the Active state. The rule is:

- If ( `grievance – risk-aversion * arrest-probability` ) > `propaganda`: become Active; else: remain Quiet.

    The `grievance`, `risk-aversion` and `arrest-probability` are determined as:

- `grievance = perceived hardship * (1 – legitimacy)`

- `risk-aversion` : fixed individual random deviate in U[0, 1]

- `arrest-probability = 1 - e^{-k*cop-ratio}`. The `cop-ratio` is the number of cops to active citizens on a cell; $k$ a constant from Epstein's model to realize a 90% `arrest-probability` if only one cop and one Active citizen occupy the same cell.

## 6.2 Controller

Nominal agent dynamics as exhibited in Epstein's model operate under a fixed values of government `legitimacy` and `propaganda`. In our control scenarios `legitimacy` is a random variate within the range (0.6, 0.85] changing at 20 random intervals over the model run.

In the control scenarios the government regulates the level of `propaganda` based on a logistic function with the state space predicted number of Active agents as the independent variable. The `propaganda` controller is specified by:

$$P(t) = \frac{P_{max} - P_{min}}{1 + exp[-m(A(t) - A_0)]} + P_{min} \tag{1}$$

where $P_{min}$ and $P_{max}$ are lower and upper bounds of `propaganda`, $m$, slope of the logistic, $A$($t$) the state space projected estimate of the number of Active citizens, and $A_0$ the Active variable midpoint of the logistic. The control function is shown in Fig 10.

**6.2.1 Stability.** To assess stability of the controller we examine the control function in the Laplace domain. The controller is defined in Eq 1 with the general logistic form $C(x) = \frac{1}{1+e^{(-x)}}$ and a Laplace transform $C(s) = \exp\left(\frac{s}{e}\right)\exp i\left(\frac{s}{e}\right)$ where expi is the exponential integral function. Evaluation in the s–domain reveals a single pole at the origin. As there is not a pole in the positive s–domain, the controller is stable. Referring to Eq 1 and Fig 10 we see controller response is bounded to a constant at large arguments, as well as bounded to a finite, non–zero value at the origin.

## 6.3 Sequential locally weighted global linear maps (s–map)

The s–map was proposed by Sugihara [9] as a method to quantify the state dependence and nonlinearity of dynamical systems, aid in the identification of chaotic systems, and provide

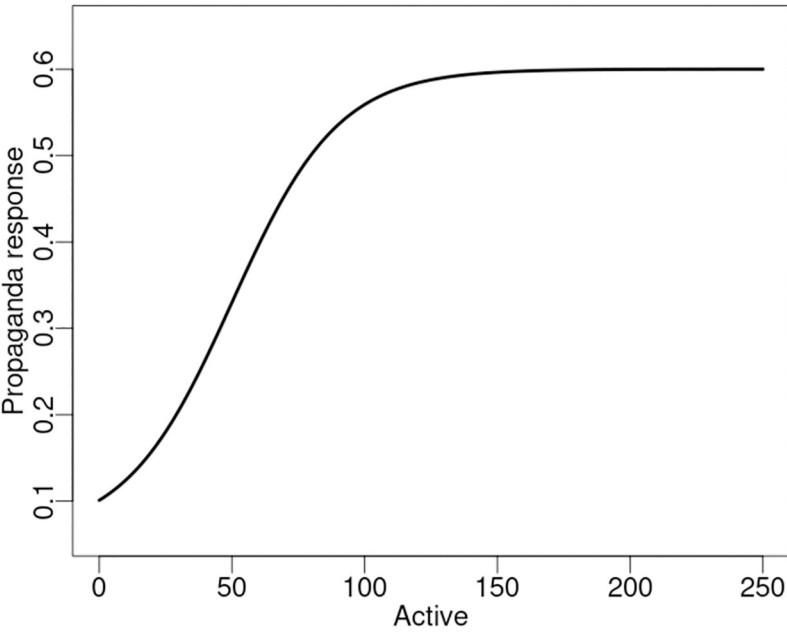

**Fig 10. Government propaganda response function (controller).** The controller operates with values: $P_{min} = 0.06$, $P_{max} = 0.6$, $m = 0.05$, $A_0 = 50$.

improved data–driven predictability of nonlinear dynamics. S–map performs linear regression on state space neighbors localized with an exponential decay kernel. The exponential localization function is $F(\theta) = \exp(-\theta d/D)$, where $d$ is the neighbor distance and $D$ the mean distance. Depending on the value of the kernel localization parameter $\theta$, neighbors close to the prediction state have a higher weight than those further from it such that a local linear approximation to the nonlinear system is reasonable. This localization can be used to identify an optimal local scale, in-effect quantifying the degree of state dependence, and therefore nonlinearity of the system. Another advantage of the localization is the potential for improved predictability as state space neighbors found with an optimal $\theta$ will best represent the state transitions.

Another feature of s–Map is that for a properly fit model, the regression coefficients between variables have been shown to approximate the gradient (directional derivative) between variables over time [39]. These Jacobians represent the time-varying interaction strengths between system variables. The s–map algorithm is detailed in algorithm 1 with notation defined in Table 2.

### 6.3.1 Notation.

**Table 2. EDM S–map notation.**

| Parameter | Description |
| --- | --- |
| $E$ | embedding dimension |
| $T_p$ | prediction horizon |
| $X \in \Re$ | observed time series |
| $y \in \Re^E$ | vector of observations |
| $\theta \geq 0$ | kernel width parameter |
| $X_t^E = (X_1, X_2, \ldots, X_{E+1})$ | state space |
| $k_N$ | row rank of $X_t^E$ |
| $\|v\|, \|v\|_2^2$ | norm of $v$, L2-norm |

**6.3.2 Nearest neighbors.** The NN method returns a list of indices $N = \{N_1, \ldots, N_k\}$ such that

$$\|X_{N_i}^E - y\| \leq \|X_{N_j}^E - y\| \text{ if } 1 \leq i \leq j \leq k,$$

**6.3.3 S–map**

**Algorithm 1** S-map

```
1: procedure SMAP(y, X, E, Tₚ, θ)
2:     N ← NN(y, X, k_N)                          ▷ Find nearest neighbors.
3:     D ← (1/k_N) Σ_{i=1}^{k_N} ‖X_{N_i}^E − y‖        ▷ Mean of distances.
4:     for i = 1, …, k do
5:         w_i ← exp(−θ‖X_{N_i}^E − y‖/D)                 ▷ Compute weights.
6:     W ← diag(w_i)                             ▷ Reweighting matrix.
```

$$
7:\quad A \leftarrow 
\begin{bmatrix}
1 & X_{N_1} & X_{N_1-1} & \cdots & X_{N_1-E+1} \\
1 & X_{N_2} & X_{N_2-1} & \cdots & X_{N_2-E+1} \\
\vdots & \vdots & \vdots & \ddots & \vdots \\
1 & X_{N_k} & X_{N_k-1} & \cdots & X_{N_k-E+1}
\end{bmatrix}
\quad \triangleright \text{Design matrix.}
$$

```
8:     A ← WA                                    ▷ Weighted design matrix.
```

$$
9:\quad b \leftarrow 
\begin{bmatrix}
X_{N_1+T_p} \\
X_{N_2+T_p} \\
\vdots \\
X_{N_k+T_p}
\end{bmatrix}
\quad \triangleright \text{Response vector.}
$$

```
10:    b ← Wb                                     ▷ Weighted response vector.
11:    ĉ ← argmin_c‖Ac − b‖²₂                     ▷ Least squares solution.
12:    ŷ ← ĉ₀ + Σ_{i=1}^{E} ĉ_i Y_i     ▷ Prediction of local linear model ĉ.
13:    return ŷ
```

## Author Contributions

**Conceptualization:** Joseph Park, George Sugihara, Gerald Pao.

**Formal analysis:** Joseph Park.

**Methodology:** Joseph Park, George Sugihara.

**Resources:** George Sugihara.

**Software:** Joseph Park.

**Supervision:** George Sugihara.

**Validation:** Joseph Park, Gerald Pao.

**Writing – original draft:** Joseph Park, George Sugihara, Gerald Pao.

**Writing – review & editing:** Joseph Park, George Sugihara, Gerald Pao.

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
