## [Decision Letter · Decision Letter 0]

24 Jan 2024

PONE-D-23-39618Control of complex systems with generalized embedding and empirical dynamic modelingPLOS ONE

Dear Dr. Joseph Park,

Thank you for submitting your manuscript to PLOS ONE. After careful consideration, we feel that it has merit but does not fully meet PLOS ONE’s publication criteria as it currently stands. Therefore, we invite you to submit a revised version of the manuscript that addresses the points raised during the review process.

We look forward to receiving your revised manuscript.

Kind regards,

Yasuko Kawahata

Academic Editor

PLOS ONE

Journal Requirements:

 Whilst you may use any professional scientific editing service of your choice, PLOS has partnered with both American Journal Experts (AJE) and Editage to provide discounted services to PLOS authors. Both organizations have experience helping authors meet PLOS guidelines and can provide language editing, translation, manuscript formatting, and figure formatting to ensure your manuscript meets our submission guidelines. To take advantage of our partnership with AJE, visit the AJE website (http://aje.com/go/plos) for a 15% discount off AJE services. To take advantage of our partnership with Editage, visit the Editage website (www.editage.com) and enter referral code PLOSEDIT for a 15% discount off Editage services. If the PLOS editorial team finds any language issues in text that either AJE or Editage has edited, the service provider will re-edit the text for free.

 A clean copy of the edited manuscript (uploaded as the new *manuscript* file).

Reviewers' comments:

Reviewer's Responses to Questions

**Comments to the Author**

1. Is the manuscript technically sound, and do the data support the conclusions?

Reviewer #1: Yes

Reviewer #2: Yes

2. Has the statistical analysis been performed appropriately and rigorously? 

Reviewer #1: N/A

Reviewer #2: N/A

3. Have the authors made all data underlying the findings in their manuscript fully available?

Reviewer #1: Yes

Reviewer #2: Yes

4. Is the manuscript presented in an intelligible fashion and written in standard English?

Reviewer #1: No

Reviewer #2: Yes

5. Review Comments to the Author

Reviewer #1: My main comments are as follows:

1) The main contribution of this paper should be highlighted. Also, compare with the existing results to reveal the superior performance.

2) An important class of complex systems is Multi-Agent Systems, which can be reviewed in Introduction part by some papers such as "https://doi.org/10.1177/095965182211056", "https://doi.org/10.1016/j.jfranklin.2021.05.033", and "10.1109/TCSII.2021.3128561".

3) The grammatical errors should be considered. Also, the overall of paper should be justified.

Reviewer #2: M/s: Control of complex systems with generalized embedding and empirical dynamic modeling

s. Generalized embedding naturally encompasses multivariate dynamics enabling state space variable cross mapping for direct assessment of multivariate contributions to the dynamics. Further, state space kernel regression allows inspection of intervariable dependencies. To illustrate this an agent based model is used to generate nonlinear dynamics which are then modeled by generalized state space embedding providing state predictions to a controller regulating the system dynamics. The method is generally applicable to any dynamic system representable in a state space.

My comments are as follows:

(1) The novelty of the work can be expressed in detail in the Abstract, Highlights, and Conclusion. The main content should concern your research idea and its experimental effect verification.

(2) The expression of the abstract should be improved. A more detailed presentation of innovation should be conducted as well as the experimental verification.

(3) Recent related work survey is currently missing. More recent literature should be cited and analyzed, such as (a) Double internal loop higher-order recurrent neural network-based adaptive control of the nonlinear dynamical system, (b) Memory Recurrent Elman Neural Network-Based Identification of Time-Delayed Nonlinear Dynamical System, (c) Modeling and adaptive control of nonlinear dynamical systems using radial basis function network, (d) A novel feed-through Elman neural network for predicting the compressive and flexural strengths of eco-friendly jarosite mixed concrete: design, simulation and a comparative study, (e) Self‐recurrent wavelet neural network–based identification and adaptive predictive control of nonlinear dynamical systems, (f) Externally Recurrent Neural Network-based identification of dynamic systems using Lyapunov stability analysis and so on.

(4) The expression logic can be improved for your proposed method so that the innovation can be clearly understood. More expression of the mathematical analysis should be conducted to show your ideas clearly. Also, please try to highlight your proposed method and focus on it.

(5) Grammar should be checked and improved for the entire content. Please try to make every sentence to be correct and easy to be understood.

(6) More experimental result comparisons with references should be conducted for advantage discussion.

(7) What about the stability of the proposed method? It is not discussed. You can refer to the following papers for your reference in the introduction and discuss the stability issue: (a)Lyapunov stability-based control and identification of nonlinear dynamical systems using adaptive dynamic programming, (b) Temporally local recurrent radial basis function network for modeling and adaptive control of nonlinear systems, (c) Diagonal recurrent neural network-based adaptive control of nonlinear dynamical systems using Lyapunov stability criterion, (d) A Lyapunov-stability-based context-layered recurrent pi-sigma neural network for the identification of nonlinear systems.

6. PLOS authors have the option to publish the peer review history of their article (what does this mean?). If published, this will include your full peer review and any attached files.

Reviewer #1: No

Reviewer #2: No

---

## [Author Response · Author response to Decision Letter 0]

8 Mar 2024

Please see the attached pdf files:

Reviewer 1 : Reviewer_1_Rev1.pdf

Reviewer 2 : Reviewer_2_Rev1.pdf

---

## [Decision Letter · Decision Letter 1]

14 May 2024

PONE-D-23-39618R1Control of complex systems with generalized embedding and empirical dynamic modelingPLOS ONE

Dear Dr. Park,

Thank you for submitting your manuscript to PLOS ONE. After careful consideration, we feel that it has merit but does not fully meet PLOS ONE’s publication criteria as it currently stands. Therefore, we invite you to submit a revised version of the manuscript that addresses the points raised during the review process.

We look forward to receiving your revised manuscript.

Kind regards,

Yasuko Kawahata

Academic Editor

PLOS ONE

Journal Requirements:

Reviewers' comments:

Reviewer's Responses to Questions

**Comments to the Author**

1. If the authors have adequately addressed your comments raised in a previous round of review and you feel that this manuscript is now acceptable for publication, you may indicate that here to bypass the “Comments to the Author” section, enter your conflict of interest statement in the “Confidential to Editor” section, and submit your "Accept" recommendation.

Reviewer #2: (No Response)

Reviewer #3: (No Response)

2. Is the manuscript technically sound, and do the data support the conclusions?

Reviewer #2: (No Response)

Reviewer #3: No

3. Has the statistical analysis been performed appropriately and rigorously? 

Reviewer #2: (No Response)

Reviewer #3: No

4. Have the authors made all data underlying the findings in their manuscript fully available?

Reviewer #2: (No Response)

Reviewer #3: No

5. Is the manuscript presented in an intelligible fashion and written in standard English?

Reviewer #2: (No Response)

Reviewer #3: No

6. Review Comments to the Author

Reviewer #2: (No Response)

Reviewer #3: The article announces the treatment of the modelling and control problem for complex nonlinear dynamical systems. However, apart from a narrative description of the significance of the targeted research topic there are no specific findings and no contribution to the nonlinear control and to nonlinear estimation area. The modeling and system identification part is missing while the control part of the article is absent too. Considering the absence of specific results the article has to be substantially rewritten after taking into account major findings and established nonlinear control and nonlinear estimation methods in the subject area.

Regarding the modelling part one would expect results on nonlinear least squares approaches or on nonlinear Kalman Filtering techniques. The modelling part is a prerequisite for the control part. In the case of model-based control precise estimation of the parameters of the dynamic model of the controlled system can be performed with the use of nonlinear Kalman Filtering or nonlinear least squares algorithms. In the case of mode-free control, identification of the dynamic model can be performed with the training of nonlinear regressors such as neural or neurofuzzy networks.

Regarding the nonlinear control of the targeted complex systems on would expect results on global linearization-based control methods or on approximate linearization-based control schemes Global linearization-based control uses transformations of the system's state variables and of the system's state-space model into equivalent linear forms where the control and estimation problem can be treated with solutions initially designed for linear systems. Approximate linearization-based control techniques considers Taylor series expansion of the system's dynamics around specific operating points through the computation of Jacobian matrices. The control and estimation problems are solved sequentially around the linearization points

7. PLOS authors have the option to publish the peer review history of their article (what does this mean?). If published, this will include your full peer review and any attached files.

Reviewer #2: No

Reviewer #3: No

---

## [Editor Report · Decision Letter 2]

30 May 2024

Control of complex systems with generalized embedding and empirical dynamic modeling

PONE-D-23-39618R2

Dear Dr. Joseph Park,

We’re pleased to inform you that your manuscript has been judged scientifically suitable for publication and will be formally accepted for publication once it meets all outstanding technical requirements.

Kind regards,

Yasuko Kawahata

Academic Editor

PLOS ONE
---

## [Editor Report · Acceptance letter]

4 Jun 2024

PONE-D-23-39618R2 

PLOS ONE

Dear Dr. Park, 

I'm pleased to inform you that your manuscript has been deemed suitable for publication in PLOS ONE. Congratulations! Your manuscript is now being handed over to our production team.

Kind regards, 

on behalf of

Dr. Yasuko Kawahata 

Academic Editor

PLOS ONE